# The Protective Role of Calbindin-D_9k_ on Endoplasmic Reticulum Stress-Induced Beta Cell Death

**DOI:** 10.3390/ijms20215317

**Published:** 2019-10-25

**Authors:** Changhwan Ahn, Eui-Man Jung, Beum-Soo An, Eui-Ju Hong, Yeong-Min Yoo, Eui-Bae Jeung

**Affiliations:** 1Laboratory of Veterinary Biochemistry and Molecular Biology, College of Veterinary Medicine, Chungbuk National University, Cheongju, Chungbuk 28644, Korea; prac@naver.com (C.A.); jemman@hanmail.net (E.-M.J.); yyeongm@hanmail.net (Y.-M.Y.); 2Department of Biomaterials Science, College of Natural Resources & Life Science, Pusan National University, Miryang 50463, Korea; anbs@pusan.ac.kr; 3College of Veterinary Medicine, Chungnam National University, 99 Daehak-ro, Suite 401Veterinary Medicine Bldg., Yuseong, Daejeon 34134, Korea; ejhong@cnu.ac.kr

**Keywords:** calbindin-D_9k_, calbindin-D_28k_, beta cells, endoplasmic reticulum

## Abstract

Intracellular calcium ion content is tightly regulated for the maintenance of cellular functions and cell survival. Calbindin-D_9k_ (CaBP-9k) is responsible for regulating the distribution of cytosolic free-calcium ions. In this study, we aimed to investigate the effect of CaBP-9k on cell survival in pancreatic beta cells. Six-month-old wildtype CaBP-9k, CaBP-28k, and CaBP-9k/28k knockout (KO) mice were used to compare the pathological phenotypes of calcium-binding protein-deleted mice. Subsequently, the endoplasmic reticulum (ER) stress reducer tauroursodeoxycholic acid (TUDCA) was administered to wildtype and CaBP-9k KO mice. In vitro assessment of the role of CaBP-9k was performed following CaBP-9k overexpression and treatment with the ER stress inducer thapsigargin. Six-month-old CaBP-9k KO mice showed reduced islet volume and up-regulation of cell death markers resulting from ER stress, which led to pancreatic beta cell death. TUDCA treatment recovered islet volume, serum insulin level, and abdominal fat storage by CaBP-9k ablation. CaBP-9k overexpression elevated insulin secretion and recovered thapsigargin-induced ER stress in the INS-1E cell line. The results of this study show that CaBP-9k can protect pancreatic beta cell survival from ER stress and contribute to glucose homeostasis, which can reduce the risk of type 1 diabetes and provide the molecular basis for calcium supplementation to diabetic patients.

## 1. Introduction

Calbindin-D_9k_ (CaBP-9k) is a 9-kDa polypeptide containing two EF-calcium-binding sites, which are usually expressed in the intestine, kidney, uterus, and pituitary gland. The major role of this protein is the buffering of intracellular calcium ions [1]. CaBP-9k regulates the amount of intracellular calcium in order to prevent cell death caused by toxic free calcium levels [2]. The intracellular calcium ion concentration is regulated via the endoplasmic reticulum (ER)-mediated pathway, in which calcium ions move across the ER membrane via calcium channels and pumps [3]. Intracellular free calcium ion and ER calcium ion levels are modulated by several calcium channels, including the sarcoplasmic reticulum calcium ion ATPase (SERCA) 2a and 2b, inositol 1,4,5-trisphosphate receptor (IP3R), and ryanodine receptor 2 (RyR2). Calcium ions are taken up into the ER by an electrogenic calcium ion pump (i.e., SERCA2a and 2b), whereas RYR2 and IP3R act on the ER membrane by opening up a calcium ion-permeable conductance to the cytoplasm [3,4]. Disruption of intracellular calcium ion homeostasis can trigger ER stress [5]. The overexpression of the calcium efflux pump’s plasma membrane calcium ion-ATPase (PMCA) has been found to deplete ER calcium storage, leading to ER stress and apoptosis [6]. Due to ER stress, the ER attempts to restore normal function by halting protein translation, degrading misfolded proteins, and increasing production of chaperones involved in protein folding [4,7]. Finally, dysregulation of the intracellular calcium ion level can result in the activation of apoptotic pathways [8,9].

Diabetes is a complex disease characterized by insulin resistance and beta-cell dysfunction [10]. In type 1 diabetes, β-cell mass is reduced by 70–80% [11]. Type 2 diabetes is characterized by more variable β-cell failure relative to insulin resistance [10]. The unfolded protein response (UPR) is induced by increased protein syntheses acting as triggers of beta cell dysfunction and apoptosis [12]. The accumulation of misfolded proteins, which aggregate in the ER lumen, causes ER stress [12]. ER stress imparts a burden on the pancreas, especially on insulin-secreting beta cells, due to the synthesis and secretion of higher amounts of insulin [5,13,14]. The ER stress triggers suppression of insulin receptor signaling and pancreatic beta cell survival [5].

Furthermore, the importance of calcium in diabetes mellitus has been shown in prospective studies, which have reported varying results regarding the association between calcium intake and risk of diabetes mellitus [15,16,17]. A meta-analysis of prospective studies previously reported an 18% decrease in the risk of incident type 2 diabetes in the highest calcium intake group (661–1200 mg/day) compared to the lowest calcium intake group (219–600 mg/day) [15]. In addition, vitamin D may influence both insulin secretion and sensitivity. The relationship between type 2 diabetes and vitamin D is based on cross-sectional and prospective studies, although a conclusive relationship has not yet been described [18,19,20]. The calcium-binding protein CaBP-9k is regulated at the transcriptional and post-transcriptional levels by 1,25-dihydroxyvitamin D3, which is the hormonal form of vitamin D [21,22]. Previously, several studies have examined the molecular regulation of active calcium transport mechanisms in order to understand the role of CaBP-9k [23,24]. Previous research has reported that CaBP-9k should be considered an important factor in active calcium transport that can be compensated for by other calcium transporter genes that maintain cellular functions in CaBP-9k knockout (KO) mice. Recently, Ahn et al., examined the expression of CaBP-9k in pancreatic beta cells and found that CaBP-9k could regulate the synthesis of insulin and insulin-secreting calcium signaling. In this study, we assessed the role of CaBP-9k in pancreatic beta cells, CaBP-9k null mice as a high fat-induced insulin resistance model, and an ER stress-alleviating agent TUDCA-fed model, and then we assessed CaBP-9k overexpression in relation to ER stress-induced cell death.

## 2. Results

### 2.1. The Effects of CaBP-9, 28k, and 9/28k Ablation on Gross Examination, Body Weight, Serological Analysis, and Insulin-Dependent Transcription Factor Expressions in the Liver

To investigate the functional role of calcium-binding proteins in the context of aging, gross examination, body weight, serological analysis, and insulin-dependent transcription factor expressions in the liver were analyzed. Only CaBP-9/28k KO mice showed significantly decreased (*p* < 0.05) weight patterns among the groups (Figure 1B). In contrast, gross examination during sacrifice found that the amounts of perigonadal adipose tissue were different (Figure 1A). Perigonadal adipose tissue covered 2/3 of the abdominal surface of WT and CaBP-28k mice, whereas CaBP-9k and 9/28k KO mice showed decreased amounts of perigonadal adipose tissue. Serum chemistry showed similar patterns for the gross examination. The serum glucose level in a fasting state did not change (Figure 1C), while the glucose level in a resting state was elevated (Figure 1D). Moreover, the insulin level was reduced (Figure 1E), and urinary glucose (Figure 1F) and water consumption (Figure 1G) were elevated, similarly to diabetic symptoms in CaBP-9k-deleted compared to WT mice. Serum lipid levels, including cholesterol, low-density lipoprotein (LDL), and high-density lipoprotein (HDL), (Figure 1H–J) also decreased through CaBP-9k ablation, even though the serum calcium level was not altered (Figure 1G). Following serological evaluation of the mice, mRNA expression levels of insulin-dependent transcription factors *Mafa* (Figure 1L), *Pck1* (Figure 1M), and *NeuroD1* (Figure 1N) in the liver were examined. These transcription factors are regulated by insulin but also expressed when demand for insulin is increased. All insulin transcription factors in the liver were upregulated upon CaBP-9k deletion. In an intraperitoneal glucose tolerance test (IPGTT) for each group, CaBP-9k ablation delayed regulation of serum glucose (Figure 1O,P). Due to the results shown in Figure 1, CaBP-9k ablation led to impairment of glucose homeostasis.

### 2.2. The Effects of CaBP-9k, 28k, and 9/28k Ablation on Pancreatic Beta Cell Death

To examine the mechanism behind the diabetic symptoms of CaBP-9k deletion mice, pancreatic tissue was stained with hematoxylin and eosin (H&E). Based on H&E-stained slides, the volume of beta cells containing pancreatic islets was evaluated by calculation. Three basic diameters could be located on 3D regular objects depending on their positions in space. Differences in islet volumes counted from those three diameters increased significantly, as the formula for volume uses a third diameter power. Islet volumes in CaBP-9k (60%), CaBP-28k (22%), and CaBP-9/28k (63%) mice were reduced compared to WT (Figure 2A,B). Next, to investigate the cause of reduced islet volume, ER stress was evaluated based on protein expression of the ER stress markers BiP, IRE1α, PERK, CHOP, and PDI using western blotting (Figure 2C,D). CaBP-9k deletion induced ER-stress marker protein expression, especially in the BiP-CHOP pathway, which can induce the caspase-3 related apoptosis pathway. CaBP-9k-deleted mice showed up-regulation of caspase-3 protein expression and increased terminal deoxynucleotidyl transferase dUTP nick end labeling (TUNEL) staining in pancreatic tissues (Figure 2E).

### 2.3. The Effects of CaBP-9k, 28k, and 9/28k Ablation on Lipid Metabolism in Adipose Tissue

Due to hypoinsulinemia resulting from ER stress-induced pancreatic beta cell death, a reduction of lipid deposition in perigonadal adipose tissue occurs. H&E-stained white adipose tissue (WAT) and brown adipose tissue (BAT) showed cell atrophy, and the weight of WAT was significantly reduced upon CaBP-9k deletion (Figure 3A,B,D,E). Furthermore, fecal lipid contents were analyzed using alkaline-methanol extraction of feces. CaBP-9k-ablated mice showed increased fecal lipid contents, which means that lipid absorption was reduced upon CaBP-9k deletion (Figure 3C,F). mRNA expression of pancreatic lipase (*Pnlip*) in the pancreas was also reduced upon CaBP-9k deletion (Figure 3G). The *Pnlip* gene is known as an insulin-responsive gene, and the result indicates that lipid metabolism was abolished by hypoinsulinemia upon pancreatic beta cell death.

Expression of the anti-appetite gene *leptin* was elevated due to CaBP-9k and 28k deletion, but no changes in dietary consumption were observed among the groups (Figure 3H,I). Expression of the preadipocyte marker *CD34* was not altered, whereas expression of the lipid marker Pparγ, which stimulates lipid uptake and adipogenesis in fat cells, was shown to be reduced using real-time PCR following CaBP-9k deletion (Figure 3J). Expression of Pparγ and fatty acid synthase was reduced upon CaBP-9k deletion (Figure 3K). These data indicate that CaBP-9k deletion can interfere with lipid accumulation in adipose tissue resulting from hypoinsulinemia caused by pancreatic beta cell death.

### 2.4. The Effects of CaBP-9k Ablation in High Fat Diet-Induced Diabetes Mellitus Model

To investigate the effect of CaBP-9k deletion on ER stress, high fat diet (HFD)-induced diabetic mice, which represent a well-known model of ER stress, were examined. Several studies have shown that oxidative stress in HFD animals may be associated with ER stress, protein degradation, and autophagy. Indeed, a prolonged HFD induced activation of ER stress. Using the HFD model, resistance to ER stress and serological changes were evaluated.

Following HFD feeding, body weight and serum glucose were elevated in WT and CaBP-9k KO mice (Figure 4A–C). Between WT and CaBP-9k KO mice, body weight was reduced in HFD-CaBP-9k KO compared to HFD-WT. In addition, perigonadal WAT accumulation and *Pnlip* mRNA expression were decreased, whereas fecal lipid contents were elevated in HFD-CaBP-9k KO compared to HFD-WT mice (Figure 4D–H). Similarly to previous data (Figure 2), the protein expression of the ER stress markers BiP, PERK, and CHOP were elevated in pancreatic beta cells (Figure 4I,J). HFD-CaBP-9k mice especially, showed synergistic up-regulation of ER stress marker protein expression compared to WT, CaBP-9k, and HFD-WT mice.

### 2.5. ER Stress Reducer Tauroursodeoxycholic Acid Reduces CaBP-9k KO-Induced ER Stress

To confirm the mechanism of CaBP-9k deletion-induced hypoinsulinemia in response to pancreatic beta cell death, tauroursodeoxycholic acid (TUDCA), which is known as an ER stress reducer, was administered to 3-month-old WT, CaBP-9k, 28k, and 9/28k KO mice for 3 months. The effects of TUDCA on recovery of CaBP-9k deletion-induced glucose tolerance and lipid metabolism were evaluated. TUDCA partly recovered reduction of WAT mass in response to CaBP-9k deletion (Figure 5A). Notably, the decreased islet volume in response to CaBP-9k deletion was recovered by TUDCA treatment (Figure 5B). The resting serum glucose (Figure 5C) and serum insulin levels (Figure 5D) were also fully recovered. In the IPGTT for each group, TUDCA recovered delayed regulation of serum glucose in CaBP-9k KO mice (Figure 1E,F). Besides, the serum cholesterol and HDL levels, which contribute to lipid deposition, were recovered by TUDCA treatment in CaBP-9k KO mice (Figure 5G,H). Furthermore, expression of ER stress markers was recovered compared to 6-month-old normal mice in Figure 2C (Figure 5I,J). However, the apoptotic marker capase-3 was not fully recovered by TUDCA treatment.

### 2.6. CaBP-9k Overexpression Recovers Insulin Secretion and Pancreatic Beta Cell Survival

To examine the role of CaBP-9k in enhancing pancreatic beta cell survival under ER stress conditions, in vitro CaBP-9k overexpression was induced in CaBP-9k-null/INS-1E pancreatic beta cells. The pcDNA3.1 vector was used to introduce CaBP-9k overexpression in INS-1E cells. Successful introduction of CaBP-9k (Figure 6A) elevated insulin secretion (Figure 6B) and cell survival (Figure 6C). The ER stress inducer thapsigargin (1 µM) was administered to induce ER stress in INS-1E cells. Elevated expression of ER stress markers, including BIP, CHOP, and PDI, was recovered in response to overexpression of CaBP-9k in INS-1E cells (Figure 6D,E). Intracellular calcium imaging using confocal microscopy showed that CaBP-9k overexpression recovered calcium homeostasis of thapsigargin-induced interference (Figure 6F).

## 3. Discussion

The correlation between vitamin D and pancreatic function has been relatively well-studied. A recent study demonstrated that vitamin D-deficient rats showed increased beta cell sensitivity to 1,25(OH)2D3-dependent insulin secretion under hypocalcemic conditions [25]. CaBP-9k is known to be a vitamin-D-dependent calcium-binding protein; however, the role of CaBP-9k in the inulin synthesis and secretion of the pancreas has not been explored.

In the previous study, 9-week-old CaBP-9k KO mice showed increased expression of ER stress markers [24]. Aging reduces the efficacy of many of these chaperones and foldases [26]. Previously, 9-week-old mice showed localization of CaBP-9k in pancreatic beta cells and reduced expression of ER chaperone proteins [24]. In the present study, deletion of CaBP-9k to induce pathological conditions was defined by measuring glucose parameters, including serum glucose level, insulin level, the expression of insulin responsive genes, and the glucose/insulin tolerance test. To verify the role of CaBP-9k in insulin-secreting beta cells, serological evaluation by measuring the serum glucose level and plasma insulin level was performed. In the absence of CaBP-9k protein, serum glucose levels were elevated, following the down-regulation of serum cholesterol (LDL and HDL), which resembles the pathologic conditions of type I diabetes mellitus. CaBP-9k KO mice showed diminished abdominal adipose deposition and increased water consumption and urine glucose levels. The glucose tolerance test result showed that the blood glucose curves of CaBP-9k KO mice moved upward compared to CaBP-9k mice. The slope indicates normal insulin action, but this delayed regulation of blood glucose in CaBP-9k KO mice was the result of a relatively low level of blood insulin compared to CaBP-9k mice. CaBP-9k, which buffers cytosolic free calcium ions, was found to be colocalized with insulin upon immunostaining. The localization of CaBP-9k to the pancreatic islets could play a role in regulation of insulin secretion or production [24]. The glucose tolerance test result, which represents insulin resistance, was not altered by CaBP-9k deletion. In turn, dysregulation of serum glucose upon CaBP-9k ablation was the result of the dysregulation of insulin production or secretion and not insulin resistance.

Decreased insulin contents in the bloodstream could have an opposite effect by promoting catabolism, especially of reserve body fat [27]. Insulin is a hormone that allows the uptake of glucose into cells for energy metabolism and blood sugar balance [28]. Insulin in the body will help to build muscle, as well as aid in the storage of fat in adipose tissues [29]. Type I diabetes, which refers to insulin-dependent diabetes, can cause weight loss [30]. The body starts burning muscle and fat for energy if it cannot gain energy from food intake. CaBP-9k KO mice showed a reduction of abdominal fat storage. Adipocyte content in WAT was reduced as tissue mass decreased. In addition, pancreas lipase (*Pnlip*) expression decreased as lipid contents in feces increased. This gene regulation may be the result of reduced blood insulin contents. In addition, the lipid-rafting factor *CD34* was not altered, whereas *Pparγ* and fatty acid synthase, which contribute to fat deposition into adipose tissue, were down-regulated by CaBP-9k deletion.

Homeostasis of the intracellular calcium ion concentration is crucial for cell survival [31]. Intracellular calcium and free-calcium ions are strictly regulated by calcium channels, calcium-binding proteins, and calcium-storing organelles [32]. The ER lumen is the major intracellular-calcium-storage compartment, and depletion of ER calcium ion content is followed by rapid accumulation inside the mitochondrial matrix through the uniporter system, resulting in ER stress and cell death [33]. Less calcium influx channels in the ER membrane represent reduced calcium concentration in the ER, which causes ER stress conditions [4,34]. Impaired ER protein trafficking upon ER stress leads to impaired pro-insulin maturation and loss of insulin content [35,36]. To study the reason for decreased serum insulin levels, the total islet volume was measured with serially sectioned tissue slides. The islet volume was decreased 0.4-fold when CaBP-9k was deleted. Based on the decrease in islet volume, cell death in pancreas islet cells was assumed. ER stress markers were upregulated in CaBP-9k-ablated mice, and increased cell death of pancreatic islets was examined by caspase-3 protein expression in pancreatic beta cells and a TUNEL assay of pancreas tissues. To evaluate the effects of CaBP-9k ablation on insulin resistance, WT and CaBP-9k KO mice were fed a HFD for 3 months. The HFD induced obesity in mice compared to a chow diet. Serum glucose and WAT levels were elevated by HFD, whereas CaBP-9k KO mice showed decreased abdominal fat deposition and fat absorption. Further, ER stress markers were synergistically elevated by HFD feeding and CaBP-9k ablation. The elevated ER stress state was recovered by the ER stress reducer TUDCA. Specifically, TUDCA treatment recovered perigonadal fat deposition and serum glucose, insulin, cholesterol, and HDL levels. These data indicate the involvement of an ER stress induced cell death pathway in the mechanism whereby TUDCA protects against cell death in the absence of CaBP-9k protein.

## 4. Materials and Methods

### 4.1. Animal Experiments

Male wild-type C57BL6 mice, weighing 25–30 g, at nine weeks of age, were obtained from Samtako (Osan, Gyeonggi, Republic of Korea). The CaBP-9k, CaBP-28k, and CaBP-9/28k KO mice were generated and bred into a congenic C57BL/6J/ 129/Ola background, as previously described [23]. All animals were housed in polycarbonate cages and acclimated in an environmentally controlled room (temperature: 23 ± 2 °C, relative humidity: 50 ± 10%, frequent ventilation, and a 12-h light/dark cycle). After approximately one week of acclimatization, the mice were divided into four groups: six months old wild-type mice (group 1), CaBP-9k KO mice (group 2), CaBP-28k KO mice (group 3), and CaBP-9/28k KO mice (group 4). Rooms were polyacryl cages with controlled oxygen concentrations (20 ± 2% O_2_). After four months (6 months old), all the mice were anesthetized by the inhalation of isoflurane for blood sample collection. After collecting the blood, mice were euthanized by cervical disclosure. Individually, the TUDCA supplementation (ad libitum feeding of standard chow supplemented with 0.4% TUDCA) experiments were set up for 3 month-old mice for 3 months. The Institutional Animal Care and Use Committee (IACUC) of Chungbuk National University approved all experimental procedures (CBNUA-1113-18-02; date: 4 May 2018)

### 4.2. Staining

The 4 μm slides of formalin-fixed, paraffin-embedded pancreas tissue was deparaffinized with xylene and rehydrated with ethanol. The tissues were embedded in paraffin, cut into sections, and stained with hematoxylin and eosin (H&E). They were captured with light microscopy (BX51; Olympus, Tokyo, Japan) and an Olympus DP controller (DP21; Olympus, Tokyo, Japan) and manager at ×200 and ×400 magnification. Beta cells containing pancreatic islet volumes were evaluated with calculation: 4/3a × 2bπ; a = radius of major axis, b = radius of minor axis).

The mice feces sample (dry weight 1 g/100 µL PBS) was stained with Sudan III (Sigma-aldrich, St. Louis, MO, USA) according to manufacturer’s instruction.

TUNEL staining was performed on paraffin-embedded sections using the In Situ Cell Death Detection Kit, Fluorescein (Roche Diagnostics, Mannheim, Germany). The slides were rinsed with PBS, and the area around the sample was dried. The slides were then incubated with 200 µL of TUNEL reaction mixture containing terminal deoxynucleotidyl transferase (TdT) for 60 min in a dark, humidified atmosphere at 37 °C. After slides were rinsed three times with PBS; they were analyzed using a fluorescence microscope (BX51 Standard Microscope, Olympus, Tokyo, Japan).

### 4.3. Collection and Serological Analysis of Serum

Blood was collected from each mouse, transferred into serum separator tubes (Microtainer tubes; Becton-Dickinson Co., New Jersey, NJ, USA), centrifuged at 400× *g* for 15 min, and aliquoted in 200 µL amounts. Serum glucose was analyzed using the glucometer AccuChek^®^ Active (Roche Diagnostics GmbH, Mannheim, Germany). The animals were fasted for 4 h before performance of blood glucose measurements. Plasma insulin level was also determined, by using the insulin ELISA kit (SHIBAYAGI, Gunma, Japan) according to the manufacturer’s instructions.

### 4.4. Fecal Lipid Quantification

An aliquot of dried feces (5 mg) was incubated in 1 mL alkaline methanol (methanol: 1 mol/L NaOH, 3:1 (vol/vol)) for 2 h at 80 °C in screw-capped tubes. After the vaporizing of all liquid in the tubes, the fecal lipids were measured with a microbalance (Sartorious, Göttingen, Germany).

### 4.5. Calculation of HOMA-IR Index

The homeostatic model assessment (HOMA) is a method used to quantify insulin resistance and beta cell function. The calculation of HOMA-IR was used formula “HOMA-IR = (glucose level (mg/dl) * insulin level (ng/mL)/405.”

### 4.6. Glucose/Insulin Tolerance Test

Intraperitoneal glucose tolerance test (IPGTT): Prior to glucose administration, mice underwent a 6 h fast to achieve a baseline blood glucose level. After the fasting time, a 2 mm distal section of the mouse’s sterilized tail was snipped using a scalpel and gently squeezed to obtain a drop of blood. The blood drop was applied directly to an AccuChek^®^ Active (Roche Diagnostics GmbH, Mannheim, Germany) to obtain a baseline T0 blood glucose reading expressed as mg/dL. Then, after a 20% glucose solution was injected (2 g of glucose/kg body mass) intraperitoneally, blood glucose levels at 30, 60, and 120 min were measured.

Intraperitoneal insulin tolerance test (IPITT): Prior to Insulin administration, mice were fasted for 4 h early in the morning (07:00). Using the same methods of IPGTT, a drop of blood was applied for a baseline T0 blood glucose reading expressed as mg/dL. The mice were given an intraperitoneal injection of insulin (0.5 U/kg). We continued to take blood samples from T0, and with blood glucose at 30, 60, and 120 min.

### 4.7. Total RNA Extraction and Quantitative Real-Time PCR

Mice were euthanized, and the pancreatic tissues rapidly exercised and washed in cold, sterile NaCl (0.9%). Total RNA was prepared with TRIzol reagent (Invitrogen, Carlsbad, CA, USA), and the concentration of RNA was determined by absorbance at 260 nm. Total RNA (1 μg) was reverse transcribed into first-strand cDNAs using Moloney murine leukemia virus (MMLV) reverse-transcriptase (iNtRON Bio, Sungnam, Gyeonggi, Korea) and random primers (9-mers; TaKaRa Bio. Inc., Otsu, Shiga, Japan). In total, 2 μL of cDNA template was added to 10 μL of SYBR Premix Ex Taq (TaKaRa Bio) and 10 pmol of each specific primer. The reactions were carried out for 40 cycles according to the following parameters: denaturation at 95 °C for 30 s, annealing at 58 °C for 30 s, and extension at 72 °C for 30 s. Fluorescence intensity was measured at the end of the extension phase of each cycle. The threshold value for the fluorescence intensity of all samples was set manually. The reaction cycle at which PCR products exceeded this fluorescence intensity threshold was identified as the threshold cycle in the exponential phase of the PCR amplification. The expressions of target genes were quantified against that of beta-actin. Relative quantification was based on a comparison to cycle threshold (CT) at a constant fluorescent intensity. The amount of transcript was inversely related to the observed CT, and for every twofold dilution in the transcript, CT was expected to increase by 1. Relative expression was calculated using the equation R = 2 ^−(Δ*C*Tsample -Δ*C*Tcontrol)^. To determine a normalized arbitrary value for target gene expression; its expression level was normalized to that of beta-actin. Extracted RNA (1 μg) was run with 1% agarose gel, and confirmed with three different rRNA bands on the gel. A reverse transcriptase (RT) negative control (without reverse transcriptase) and a non-template negative control were included to confirm the absence of genomic DNA or a contamination in the reactions. Each sample contained three biological replicates and each biological replicate had the technical triplicates. All methods and data collections were confirmed to follow the MIQE (Minimum Information for the Publication of Quantitative Real Time PCR Experiments) guidelines [37].

### 4.8. Western Blot Analysis

The pancreases of euthanized mice were rapidly excised and washed in a cold, sterile 0.9% NaCl solution. Protein was extracted with Pro-prep (InTron., Inc., Seoul, Korea) according to the manufacturer’s instructions. Protein (50 mg per lane) was separated with 10% SDS-polyacrylamide gel electrophoresis (PAGE) and transferred to a polyvinylidene fluoride transfer membrane (Perkin Elmer Co., Wellesley, MA, USA) in a TransBlot Cell (TE-22, Hoefer Co., San Francisco, CA, USA) according to the manufacturer’s protocol. The resulting blot was blocked in TBS-T containing 5% skim milk for 60 min, then incubated in target primary antibody or beta-actin (rabbit-monoclonal, 1:2000, Assay Design, Inc., Ann Arbor, MI, USA) for 4 h at room temperature. After washing in buffer, the membranes were incubated with the appropriate horseradish peroxidase-conjugated secondary antibodies (anti-rabbit, 1:2000, Santa Cruz, Santa Cruz, CA, USA or anti-mouse, 1:5000, Santa Cruz) for 1 h at room temperature (RT). After washing, the blots were developed by incubation in ECL chemiluminescence reagent (Santa Cruz), and were subsequently exposed to Biomax Light film (Kodak, New York, NY, USA) for 1–5 min. Signal specificity was confirmed by blotting in the absence of primary antibody, and bands were normalized to beta-actin immunoreactive bands visualized in the same membrane after stripping. Density measurements for each band were performed with NIH Image J software. Background samples from an area near each lane were subtracted from each band to obtain mean band density.

### 4.9. Cell Culture

Rat insulinoma beta cell line INS-1E was purchased from the Korean Cell Line Bank (Seoul, South Korea). INS-1E cells were incubated at 37 °C; all cell culture materials were obtained from Invitrogen Life Technologies (Carlsbad, CA, USA). Transient CaBP-9k overexpressing INS-1E cells were generated as described previously, using Lipofectamine for transfection and G418 (Invitrogen, Carlsbad, CA, USA) for selection. Cells were cultured in RPMI 1640 (Invitrogen) containing 11.1 mM glucose and supplemented with 10% fetal bovine serum, 1% penicillin-streptomycin, 1 mM sodium pyruvate, 2 mM l-glutamine, 10 mM HEPES, and 0.05 mM 2-mercaptoethanol. In total, 1 μmol/mL of thapsigargin was used to induce ER stress.

### 4.10. Glucose Stimulated Insulin Secretion Measurement

INS-1E cells were preincubated in 0.5% (wt/vol) bovine serum albumin, HEPES-buffered saline in 2.8 mM glucose at 37 °C in 5% CO_2_ for 30 min and then transferred to 0.5% bovine serum albumin, HEPES-buffered saline in 2.8 mM glucose, with 2 mM—stimulatory—CaCl_2_ alone. After incubation at 37 °C in 5% CO_2_ for 30 min, the supernatants were measured for insulin release with Insulin ELISA kit (SHIBAYAGI, Gunma, Japan).

### 4.11. Cell Proliferation Assay

INS-1E cells were seeded at a density of 0.5 × 10^3^ cells per well in 96-well plates, and 100 μL of media was used for cell culture—DMEM high-glucose media (Biowest, Nuaillé, France). EZ-Cytox enhanced cell viability assay reagent (DoGenBio, Seoul, Korea) at the recommended concentration, was then placed in each well. One hour later, the absorbance value at 450 nm was measured with an Epoch microplate spectrophotometer (BioTek, Winooski, VT, USA). Cell viability (%) was determined by comparing optical density (OD) values via the formula ODsample/ODcontrol × 100, for each concentration range. A cell survival curve was calculated from the values obtained, and the cell survival rate was calculated.

### 4.12. Confocal Microscopy

Ca^2+^ imaging was performed using a confocal laser scanning microscope (Zeiss, LSM 710). INS-1E cells (3 × 10^3^ cells) were attached onto a cover-glass bottom dish (SPL 100350) in medium for 12 h. INS-1E cells were loaded with fluo-4-AM (5 μM; Invitrogen) for 30 min at 37 °C, then transferred to Ca^2+^-free BSS buffer solution (5.4 mM KCl, 5.5 mM d-glucose, 1 mM MgSO_4_, 130 mM NaCl, and 20 mM HEPES at pH 7.4). Cells were treated with CaCl_2_ (2 mM) on the holder in the first phase for measurement of the inflow rate of calcium in cytosol. Detection of calcium imaging was performed at 3 s intervals for 20 min using a confocal microscope.

### 4.13. Statistical Analysis

The results of all experiments are presented as the means ± SDs. The number of mice for each group was eight. Data were analyzed with a nonparametric, one-way analysis of variance (ANOVA), using Tukey’s test for multiple comparisons, and non-parametric two-way ANOVA. Data were ranked according to those tests. For power and sample size calculations, the variations of values for given settings of false positive rates (α) and power (1-β) were used. The power is the probability of failing to detect a difference when there is an actual difference. All statistical analyses were performed using Graphpad^TM^ software. *p* < 0.05 was considered statistically significant.

## 5. Conclusions

CaBP-9k plays a protective role on pancreatic beta cells by ablating cellular CaBP-9k protein. CaBP-9k KO mice showed ER stress-induced cell death resulting from the dysregulation of intracellular calcium concentration. Cell death induces a reduction of insulin and inhibits anabolism similarly to normal lipid deposition. The CaBP-9k mutation may be one of the predictive factors for high-risk individuals in type 1 diabetes.

## Figures and Tables

**Figure 1 ijms-20-05317-f001:**
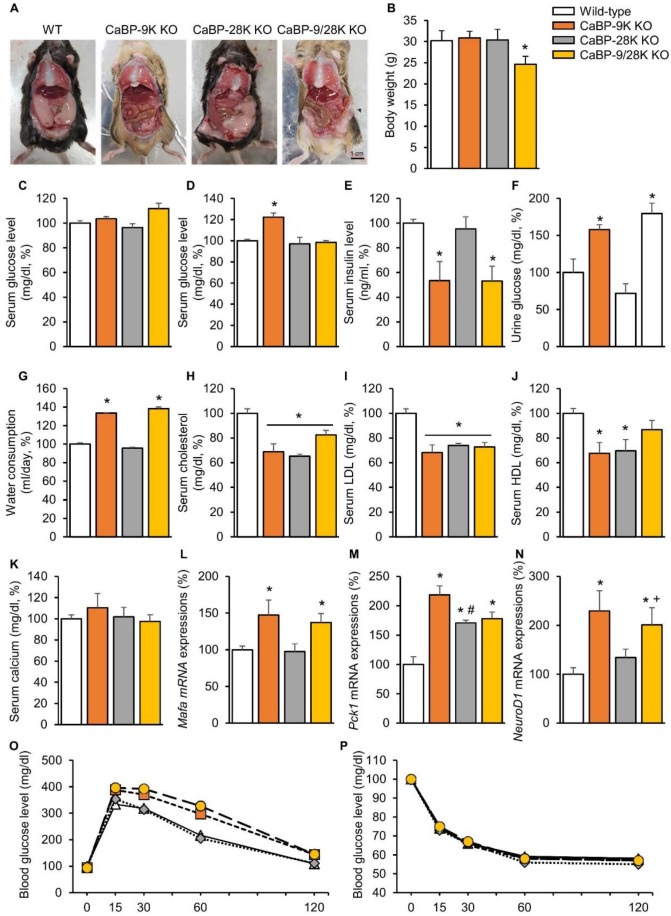
The effects of CaBP-9k, 28k, and 9/28k ablation on glucose metabolism. Abdominal adipose tissue deposition in mice (**A**), body weights of mice (**B**), serum glucose level in a fasting state (**C**), serum glucose level at resting state (**D**), insulin level (**E**), urine glucose level (**F**), water consumption (**G**), serum cholesterol (**H**), serum low-density lipoprotein (LDL) level (**I**), serum high-density lipoprotein (HDL) level (**J**), serum calcium level (**K**), insulin dependent transcription factor in liver *Mafa* (**L**), *Pck1* (**M**), *NeuroD1* (**N**), mRNA expression and glucose tolerance test (**O**), and insulin tolerant test result (**P**). Values are expressed as means ± SDs; * *p* < 0.05 versus WT; # *p* < 0.05 versus CaBP-9k knockout (KO); + *p* < 0.05 versus CaBP-28k KO. Scale bar = 1 cm.

**Figure 2 ijms-20-05317-f002:**
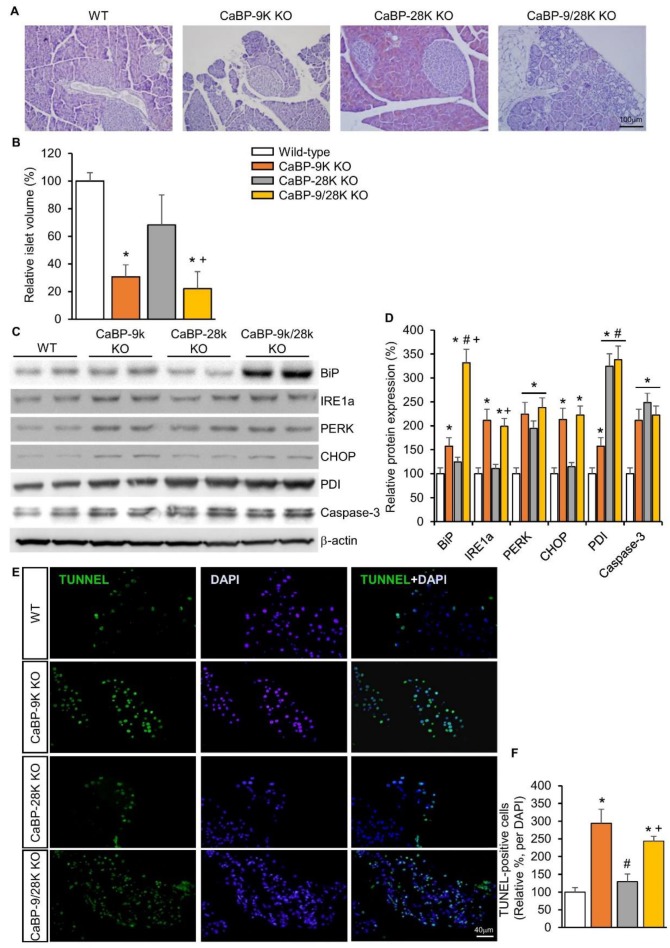
The effects of CaBP-9k, 28k, and 9/28k ablation in pancreatic beta cell death. hematoxylin and eosin (H&E) staining in pancreatic tissue (**A**); assessment of beta cell mass in pancreatic tissue (**B**); endoplasmic reticulum stress marker protein expression in pancreas (**C**,**D**); TUNNEL assay with pancreatic tissue (**E**,**F**). Values are expressed means ± SDs; * *p* < 0.05 versus WT; # *p* < 0.05 versus CaBP-9k KO; + *p* < 0.05 versus CaBP-28k KO. Scale bar, 100 µm in (**A**); 40 µm in (**E**).

**Figure 3 ijms-20-05317-f003:**
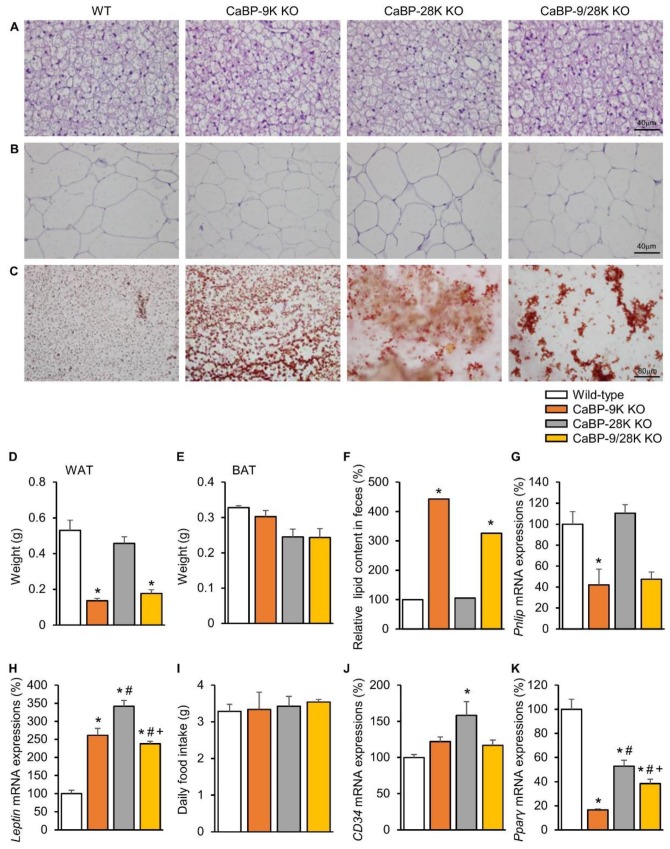
The effects of CaBP-9k, 28k, and 9/28k ablation in lipid metabolism in adipose tissue. Lipid metabolism analysis with brown adipose tissue (BAT) (**A**) and white adipose tissue (WAT) (**B**) histology using H&E staining. Fecal lipid staining with Sudan III (**C**). Measurement of WAT (**D**) and BAT (**E**) weights for each group. Lipid contents in fecal quantification (**F**) and the expression of pancreatic lipase expression (*Pnlip*) in pancreatic tissue (**G**) were examined. *Leptin* (**H**), daily food intake (**I**), *CD34* (**J**), and *Ppar*γ (**K**) regulate lipid deposition in adipose tissue. Values are expressed means ± SDs; * *p* < 0.05 versus WT; # *p* < 0.05 versus CaBP-9k KO; + *p* < 0.05 versus CaBP-28k KO. Scale bar, 40 µm in (**A**,**B**); 80 µm in (**C**).

**Figure 4 ijms-20-05317-f004:**
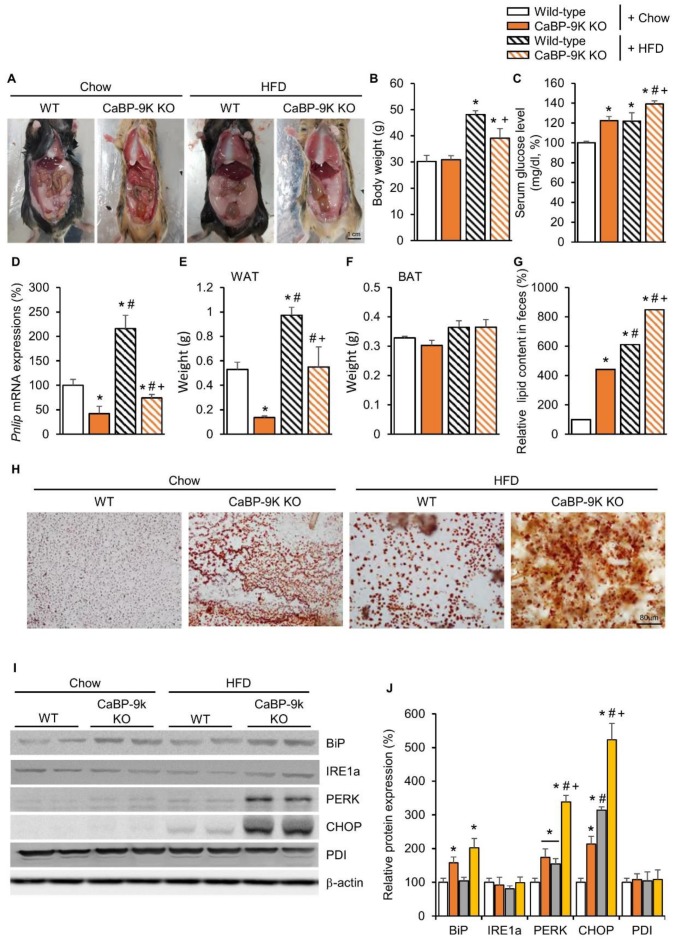
The effects of CaBP-9k ablation in high-fat diet (HFD)-induced diabetes mellitus model. Lipid deposition for each group (**A**); body weight (**B**), serum glucose level at resting state (**C**), pancreatic lipase expression (**D**), WAT (**E**) and BAT (**F**) mass, and fecal lipid content (**G**); and fecal sudan III staining (**H**). ER stress marker protein expression in pancreas (**I**,**J**). Values are expressed means ± SDs * *p* < 0.05 versus WT chow diet; # *p* < 0.05 versus CaBP-9k KO chow diet; + *p* < 0.05 versus WT HFD. Scale bar = 1 cm

**Figure 5 ijms-20-05317-f005:**
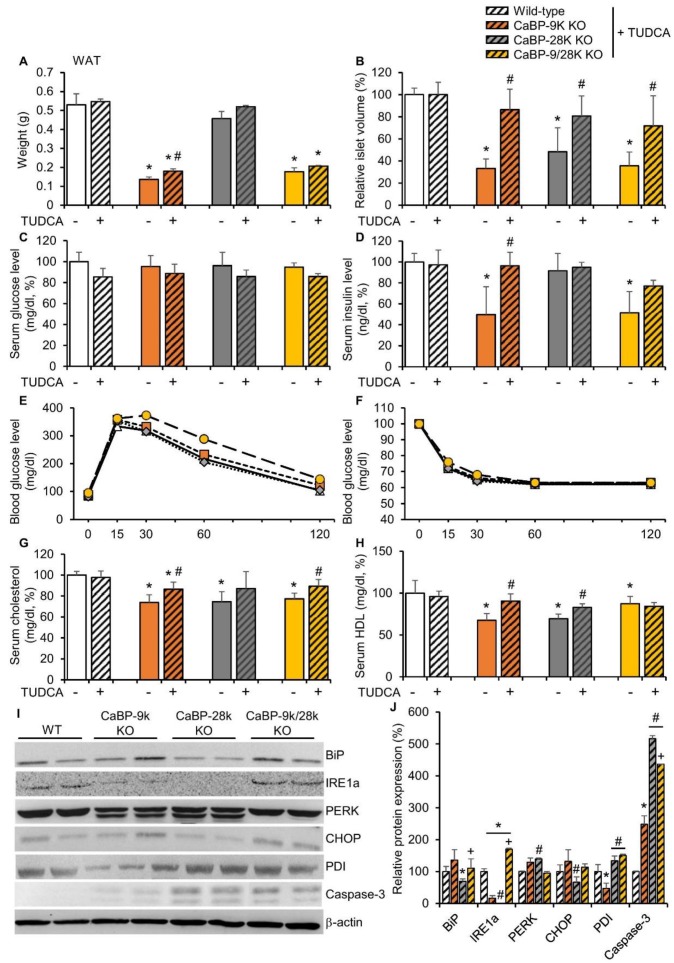
Tauroursodeoxycholic acid reduced CaBP-9k KO induced ER stress. TUDCA treatment recovered CaBP-9k deletion induced glucose tolerance and malregulation of lipid metabolism. The WAT mass (**A**), islet volume (**B**), serum glucose level at resting state (**C**), serum insulin level (**D**), glucose tolerance test (**E**), insulin tolerance test (**F**), serum cholesterol level (**G**), and serum HDL level (**H**) were recovered in response to TUDCA treatment. * *p* < 0.05 versus WT normal diet; # *p* < 0.05 versus normal diet in same mice genotype. ER stress marker protein expression in pancreas (**I**,**J**). Values are expressed means ± SDs; * *p* < 0.05 versus WT; # *p* < 0.05 versus CaBP-9k KO; + *p* < 0.05 versus CaBP-28k KO.

**Figure 6 ijms-20-05317-f006:**
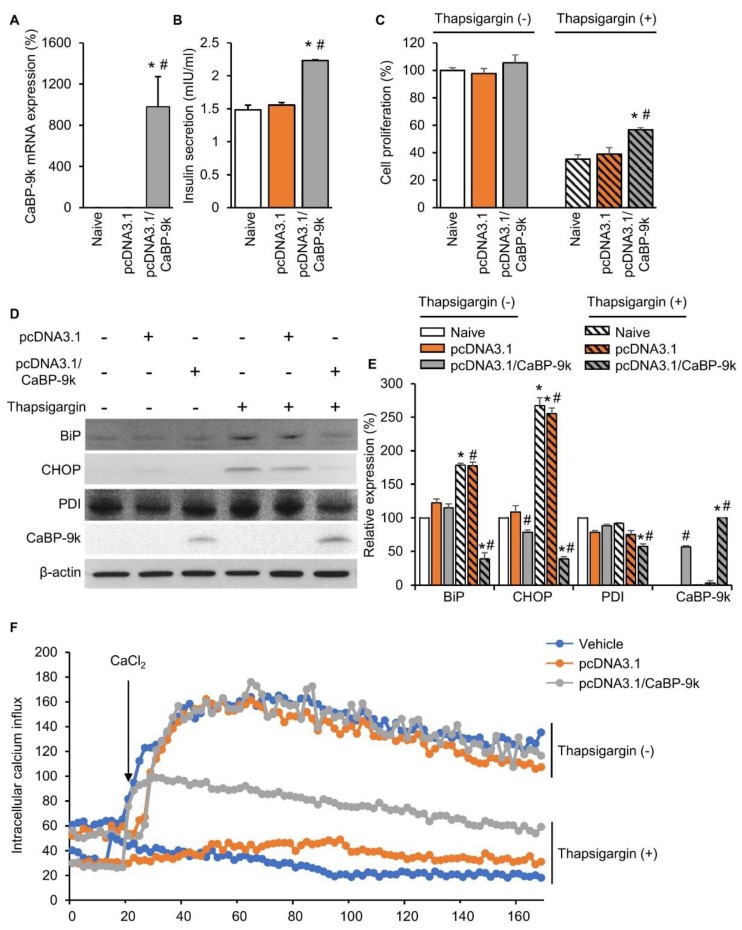
CaBP-9k overexpression recovered insulin secretion and pancreatic beta cell survival. The CaBP-9k expression was successfully induced by overexpression vector pcDNA 3.1(+) (**A**). Overexpressed CaBP-9k increased insulin secretion (**B**), cell survival rate (**C**), and alleviation of ER stress marker expression (**D**,**E**). Intracellular calcium influx measurement with confocal microscopy (**F**). Values are expressed means ± SDs; * *p* < 0.05 versus Naive; # *p* < 0.05 versus pcDNA3.1.

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
