# Peer review of "The Protective Role of Calbindin-D9k on Endoplasmic Reticulum Stress-Induced Beta Cell Death"

_ijms, 2019, doi:10.3390/ijms20215317_

Round 1

Reviewer 1 Report

In this manuscript, the authors showed that the role of calbindin-D9k in beta cell against endoplasmic reticulum stress induced cell death. The significant findings from this study were that CaBP-9k play a protective effect for pancreatic beta cell by ablating cellular CaBP-9k protein. Furthermore, the authors found that the CaBP-9k protects pancreatic beta cell survival from ER stress which contribute to glucose homeostasis which can reduce the risk of type 1 diabetes and molecular basis for calcium supplementation to diabetic patients. The topic of the paper is interesting although some minor concerns have been raised after carefully reading the work.

1.      The authors need to put scale bar in Figures.

2.      The authors need to follow the MIQE-approved term (Reverse Transcription - quantitative PCR (RT-qPCR)).

3.      The authors need to use absolute quantification for relative quantification rather than relative % for better understanding.

Author Response

Response to reviewers’ comments

We would like to thank the reviewer for constructive suggestions to improve the quality of this manuscript. In the revised version, we have addressed the concerns that the reviewer raised.

REVIEWER 1

In this manuscript, the authors showed that the role of calbindin-D9k in beta cell against endoplasmic reticulum stress induced cell death. The significant findings from this study were that CaBP-9k play a protective effect for pancreatic beta cell by ablating cellular CaBP-9k protein. Furthermore, the authors found that the CaBP-9k protects pancreatic beta cell survival from ER stress which contribute to glucose homeostasis which can reduce the risk of type 1 diabetes and molecular basis for calcium supplementation to diabetic patients. The topic of the paper is interesting although some minor concerns have been raised after carefully reading the work.

The authors need to put scale bar in Figures.

Response: We appreciate the reviewer for the positive comments. Following the reviewer’s comment, we added scale bar in the Figure 1A and Figure 4A.

The authors need to follow the MIQE-approved term (Reverse Transcription - quantitative PCR (RT-qPCR)).

Response: Following the reviewer’s comment, we added statement following MIQE guideline.

Line 312-318: “Extracted RNA (1 μg) was run with 1% agarose gel, and confirm with three different rRNA band with gel. An RT negative control (without reverse transcriptase) and a non-template negative control were included to confirm the absence of genomic DNA or contamination in the reactions. Each sample contained three biological replicates and each biological replicate had the technical triplicates. All methods and data collections were confirmed to follow the MIQE (Minimum Information for Publication of Quantitative Real Time PCR Experiments) guidelines [36].”

The authors need to use absolute quantification for relative quantification rather than relative % for better understanding.

Response: Following the reviewer’s comment, we changed graphs in the Figure 1C-K, 4C, 5C, 5D, 5G and 5H.

"Please check attached files"

Reviewer 2 Report

This manuscript of Ahn and colleagues is designed to look at the effect of vitamin-D-dependent calcium-binding protein (CaBP-9k) on ER induced stress in the beta cell of CaBP-9k knockout mouse model.

The study is interesting. Although the authors had carried out many experiments, the manuscript failed to convey sound conclusions in the current format. I will outline briefly the reasons behind such failure.

1- Conceptual: The authors require to put a clear hypothesis and aligning the interpretation and conclusions to answer the main research questions of the proposed hypothesis.

a.       It is not clear whether the authors wanted to discuss the protective effect of CaBP-9k in type 1 diabetes or type 2? I experienced mixed ideas and thoughts within this manuscript.

b.       The authors reported decline in islets size and assumed beta cell death within the CaBP-9k knockout mouse. However, this does not imply that this is a genuine model of type 1 diabetes. In addition, they claimed that treatment with TUDCA recovered islet volume and beta cell function. How this happened if they described what happened as cell death.

c.       Line: 131. Effects of CaBP-9k ablation in high fat diet induced diabetes mellitus model. They should specify what type of model? Does the authors want to report CaBP-9k knockout mouse as a model for Type 1 diabetes?  Is this a genuine model? They require to make these points clearer in the manuscript.

Ok, I did little search, it seems that the authors reported in 2016 CaBP-9k knockout mouse as a model of type 1 diabetes (Ref22). You need to clearly address your research questions, and reflect adequately on your previous related work.

 Line 71: Recently, Ahn et al., examined expression of CaBP-9k in pancreatic beta cell.  In this study, we assessed the role of CaBP-9k in pancreatic beta cell survival with CaBP-9k KO mice  and CaBP-9k overexpression related to ER stress induced cell death.

This is not sufficient information; you need to be more informative when you cite your work especially when you want to address the key research questions.

d.       Line: 54: Diabetes is a complex disease characterized by both insulin resistance and beta-cell dysfunction 54 [10]: Is Ref 10 the best one to cite this general statement?  there are major papers that could be cited, and your reference list is small(34 only). Again, you need to be more specific, these are type 2 diabetes features.  

e.       Line: 55-56: ER stress causes a burden to the pancreas, especially on insulin-secreting beta cells, owing to the synthesis and secretion of higher amounts of insulin [5, 11, 12]. This burden creates ER stress, which triggers suppression of insulin receptor signaling and pancreatic beta cell survival [5]. How ER stress causes a burden AND This burden creates ER stress? The whole sentence needs to be rephrased.

f.        Conclusion: Only the first few lines talked about the study, the others are discussion points that not suited for the conclusion. Additionally, the presentation of the conclusion after methodology is weird; I assume this a technical error.

2-Technical: The figures, which display data presented in this manuscript, are absent. My judgement are therefore based on the data presented in the text. There are also other technical issues.

a.       English, I appreciate that this came from non-native speaker, but the language of the manuscript requires some improvement. This will help to make the ideas and conclusion sharper.

b.       The authors require being more rational in data presentation. Example: line 75: the first paragraph of discussion was not at all part of the result (6-7 lines). These require to be relocated either to methods or to be used in discussion.  

c.       Other minors: Grammar and Typos: some examples: glucose tolerant test was used repeatedly in the manuscript. It should be “glucose tolerance test”. Line 84: were quite differ by… the language of the conclusions was very poor: Line369: The people who has mutated CaBP-9k are might highly susceptible to have diabetes due to this research.

Author Response

Response to reviewers’ comments

We would like to thank the reviewer for constructive suggestions to improve the quality of this manuscript. In the revised version, we have addressed the concerns that the reviewer raised.

REVIEWER 2

This manuscript of Ahn and colleagues is designed to look at the effect of vitamin-D-dependent calcium-binding protein (CaBP-9k) on ER induced stress in the beta cell of CaBP-9k knockout mouse model.

The study is interesting. Although the authors had carried out many experiments, the manuscript failed to convey sound conclusions in the current format. I will outline briefly the reasons behind such failure.

1- Conceptual: The authors require to put a clear hypothesis and aligning the interpretation and conclusions to answer the main research questions of the proposed hypothesis.

It is not clear whether the authors wanted to discuss the protective effect of CaBP-9k in type 1 diabetes or type 2? I experienced mixed ideas and thoughts within this manuscript.

Response: We appreciate the reviewer for the positive comments. Throughout the manuscript, the CaBP-9k have protective effect against type 1 diabetes which is originated from pancreatic beta cell death. In Figure 4, we tried to verify relationship between CaBP-9k deletion effect and insulin resistance. High fat diet is well known material to induce the insulin resistance in mice. With this insulin resistance model, we found that ER stress which can induce both the cell death and the insulin resistance. Therefore, the primary hypothesis is, like the reviewer pointed out, the protective effect of CaBP-9k in type 1 diabetes. To avoid the confusion, we change the expression from “in type II diabetes” to “with insulin resistance” in the end of discussion section (Line 222).

The authors reported decline in islets size and assumed beta cell death within the CaBP-9k knockout mouse. However, this does not imply that this is a genuine model of type 1 diabetes. In addition, they claimed that treatment with TUDCA recovered islet volume and beta cell function. How this happened if they described what happened as cell death.

Response: We thought that the effect of CaBP-9k deletion alone is not critical for the vitality of the mice. It is been a two decades for study with CaBP-9k null mice. We regard the effect of CaBP-9k deletion was partially compensated with other calcium channels, buffering proteins and transporters (We examined in 2007, Lee GS et al, Phenotype of a calbindin-D9k gene knockout is compensated for by the induction of other calcium transporter genes in a mouse model. J Bone Miner Res). The TUDCA is ER stress reducer, so it recovered the CaBP-9k deletion effect at the latter part of cell death mechanism with ER stress.

Line: 131. Effects of CaBP-9k ablation in high fat diet induced diabetes mellitus model. They should specify what type of model? Does the authors want to report CaBP-9k knockout mouse as a model for Type 1 diabetes? Is this a genuine model? They require to make these points clearer in the manuscript.

Ok, I did little search, it seems that the authors reported in 2016 CaBP-9k knockout mouse as a model of type 1 diabetes (Ref22). You need to clearly address your research questions, and reflect adequately on your previous related work.

Response: As we described for question a., we tried to verify relationship between CaBP-9k deletion effect and insulin resistance. High fat diet is well known material to induce the insulin resistance in mice. With this insulin resistance model, we found that ER stress which can induce both the cell death and the insulin resistance. Therefore, the primary hypothesis is, like the reviewer pointed out, the protective effect of CaBP-9k in type 1 diabetes.

Line 71: Recently, Ahn et al., examined expression of CaBP-9k in pancreatic beta cell.  In this study, we assessed the role of CaBP-9k in pancreatic beta cell survival with CaBP-9k KO mice and CaBP-9k overexpression related to ER stress induced cell death.

This is not sufficient information; you need to be more informative when you cite your work especially when you want to address the key research questions.

Response: Following the reviewer’s comment, we changed the description.

Line 75-79: “Recently, Ahn et al., examined the expression of CaBP-9k in pancreatic beta cells and found that CaBP-9k could regulate the synthesis of insulin and insulin-secreting calcium signaling. In this study, we assessed the role of CaBP-9k in pancreatic beta cells, CaBP-9k null mice as a high fat-induced insulin resistance model, and an ER stress-alleviating agent TUDCA-fed model, and then we assessed CaBP-9k overexpression in relation to ER stress-induced cell death.”

Line: 54: Diabetes is a complex disease characterized by both insulin resistance and beta-cell dysfunction 54 [10]: Is Ref 10 the best one to cite this general statement? there are major papers that could be cited, and your reference list is small (34 only). Again, you need to be more specific, these are type 2 diabetes features.

Response: Following the reviewer’s comment, we changed the phase and references.

Line 54-56: “Diabetes is a complex disease characterized by insulin resistance and beta-cell dysfunction [10]. In type 1 diabetes, β-cell mass is reduced by 70–80% [11]. Type 2 diabetes is characterized by more variable β-cell failure relative to insulin resistance [10].”

Line: 55-56: ER stress causes a burden to the pancreas, especially on insulin-secreting beta cells, owing to the synthesis and secretion of higher amounts of insulin [5, 11, 12]. This burden creates ER stress, which triggers suppression of insulin receptor signaling and pancreatic beta cell survival [5]. How ER stress causes a burden AND This burden creates ER stress? The whole sentence needs to be rephrased.

Response: Following the reviewer’s comment, we added the descriptions and references.

Line 54-58: “Diabetes is a complex disease characterized by insulin resistance and beta-cell dysfunction [10]. In type 1 diabetes, β-cell mass is reduced by 70–80% [11]. Type 2 diabetes is characterized by more variable β-cell failure relative to insulin resistance [10]. The unfolded protein response (UPR) is induced by increased protein synthesis acting as triggers of beta cell dysfunction and apoptosis [12]. The accumulation of misfolded proteins, which aggregate in the ER lumen, causes ER stress [12].”

Conclusion: Only the first few lines talked about the study, the others are discussion points that not suited for the conclusion. Additionally, the presentation of the conclusion after methodology is weird; I assume this a technical error.

Response: Following the reviewer’s comment, we deleted not suitable region of the conclusion section.

The order is due to the instruction to author from MDPI guideline.

Line 375-379: “CaBP-9k plays a protective effect on pancreatic beta cells by ablating cellular CaBP-9k protein. CaBP-9k KO mice showed ER stress-induced cell death resulting from dysregulation of intracellular calcium concentration. Cell death induces reduction of insulin and inhibits anabolism similar to normal lipid deposition. People with CaBP-9k mutation might be highly susceptible to diabetes. These results could be used for risk assessment in potential diabetic patients.”

2-Technical: The figures, which display data presented in this manuscript, are absent. My judgement are therefore based on the data presented in the text. There are also other technical issues.

English, I appreciate that this came from non-native speaker, but the language of the manuscript requires some improvement. This will help to make the ideas and conclusion sharper.

Response: Following the reviewer’s comment, we get the secondary English correction of all parts from the English editing service team.

The authors require being more rational in data presentation. Example: line 75: the first paragraph of discussion was not at all part of the result (6-7 lines). These require to be relocated either to methods or to be used in discussion.

Response: Following the reviewer’s comment, we relocated the phase to the discussion part

Line 178-181: “In the previous study, 9-week-old CaBP-9k KO mice showed increased expression of ER stress markers. Aging reduces the efficacy of many of these chaperones and foldases. Previously, 9-week-old mice showed localization of CaBP-9k in pancreatic beta cells and reduced expression of ER chaperone proteins [24].”

Other minors: Grammar and Typos: some examples: glucose tolerant test was used repeatedly in the manuscript. It should be “glucose tolerance test”. Line 84: were quite differ by… the language of the conclusions was very poor: Line369: The people who has mutated CaBP-9k are might highly susceptible to have diabetes due to this research.

Response: Following the reviewer’s comment, we fixed typo- and grammatical errors

Line 188, 194, 483 and 506: Change glucose tolerant test to glucose tolerance test

Line 85-86: “In contrast, gross examination during sacrifice found that the amounts of perigonadal adipose tissue were different (Figure 1A)”

Change conclusion part description

Line 375-379: “CaBP-9k plays a protective effect on pancreatic beta cells by ablating cellular CaBP-9k protein. CaBP-9k KO mice showed ER stress-induced cell death resulting from dysregulation of intracellular calcium concentration. Cell death induces reduction of insulin and inhibits anabolism similar to normal lipid deposition. People with CaBP-9k mutation might be highly susceptible to diabetes. These results could be used for risk assessment in potential diabetic patients.”

"Please check attached files"

Round 2

Reviewer 2 Report

The manuscript has improved but there ae sill some issues remain to be sortred.

1-Line58-61: ER stress imparts a burden on…… this burden creates….. pancreatic beta cell survival: This whole sentence is redundant now after inserting the new paragraph.

2-Line 81-84: First the title as effects on glucose metabolism not representing the whole data, the authors presented other data including weight, adipose tissues, lipid profile etc.

The first few sentences are not results; this is either methods or discussion.

3-lines 178-179: in the previous study, 9 weeks old….. aging reduces…. . Reference for this statement is required.

4-Line 367: the results of all experiments were presented as MEAN+/- SD. Then all data were analysed in non-parametric statistics? Is your data not normally distributed?  Why to present data as mean if not?

5-Line 371: you mentioned for power and sample calculation, variation of …..? this is statement is not complete, there is part of the sentence missing?

6-line 378-379: conclusion: the last sentence is not satisfactory: the authors experimented in animal model, then they made a wide conclusion on human: people with….. might be highly susceptible to…… this could be used in a discussion but not a conclusion from the study. In addition, you need to be specific, which diabetes you are talking about here, type 1 or type 2?

7-all figures have the figure name appeared both on the top and upper part of the figure, omit the upper title.

8- In all figures, all statistical differences were represented alphabetically which is misleading in such complex figures, it will look better by using standard symbols like *, **, *** etc.

9- If possible write the under each scale on figure the size of the scale.

10-on figure 6c: you put a line on the top of three bars and denoted a which p<0.05 vs naïve, why the line covers the three bars then?

11- the title overall could be improved: The protective role of Calbindin-D9k on endoplasmic reticulum induced beta cell death.

Author Response

Response to reviewers’ 2nd comments

We would like to thank the reviewer for the second suggestion to improve the quality of this manuscript. In the revised version, we have addressed the concerns that the reviewer raised.

The manuscript has improved but there ae sill some issues remain to be sortred.

1-Line58-61: ER stress imparts a burden on…… this burden creates….. pancreatic beta cell survival: This whole sentence is redundant now after inserting the new paragraph.

Response: Following the reviewer’s comment, we changed the description with avoiding redundancy.

Line 60-63: “ER stress imparts a burden to the pancreas, especially on insulin-secreting beta cells, due to the synthesis and secretion of higher amounts of insulin [5,13,14]. The ER stress triggers suppression of insulin receptor signaling and pancreatic beta cell survival [5].”

2-Line 81-84: First the title as effects on glucose metabolism not representing the whole data, the authors presented other data including weight, adipose tissues, lipid profile etc.

Response: Following the reviewer’s comment, we changed the subtitle from “Effects of CaBP-9k, 28k, and 9/28k ablation on glucose metabolism” to “Effects of CaBP-9, 28k, and 9/28k ablation on gross examination, body weight, serological analysis and insulin-dependent transcription factor expressions in liver” (Line 83-84)

The first few sentences are not results; this is either methods or discussion.

Response: Following the reviewer’s comment, we changed the description of the first few sentences.

Line 85-87: “To investigate the functional role of calcium-binding proteins in the context of aging, gross examination, body weight, serological analysis and insulin-dependent transcription factor expressions in liver were analyzed.”

3-lines 178-179: in the previous study, 9 weeks old….. aging reduces…. . Reference for this statement is required.

Response: Following the reviewer’s comment, we added references for those statements.

Line 189-190: Ahn, C.; Lee, D.; Lee, J.H.; Yang, H.; An, B.S.; Jeung, E.B. Calbindin-D9k Ablation Disrupt Glucose/Pancreatic Insulin Homeostasis. PLoS One 2016, 11, e0164527, doi:10.1371/journal.pone.0164527.

Brown, M.K.; Naidoo, N. The endoplasmic reticulum stress response in aging and age-related diseases. Front Physiol 2012, 3, 263, doi:10.3389/fphys.2012.00263.

4-Line 367: the results of all experiments were presented as MEAN+/- SD. Then all data were analysed in non-parametric statistics? Is your data not normally distributed? Why to present data as mean if not?

Response: According to the Central Limit Theorem, when the sample size is larger than 30, the samples can be used for parametric tests. If the sample size does not meet the guidelines for the parametric tests, a nonparametric test should be used. the sample size more than 30 Nonparametric tests have less power to begin.

5-Line 371: you mentioned for power and sample calculation, variation of …..? this is statement is not complete, there is part of the sentence missing?

Response: Following the reviewer’s comment, we added description for power calculation.

Line 398-400: “For power and sample size calculation, variation of value for a given settings of false positive rates (α) and power (1-β). The power is the probability of failing to detect a difference when actually there is a difference.”

6-line 378-379: conclusion: the last sentence is not satisfactory: the authors experimented in animal model, then they made a wide conclusion on human: people with….. might be highly susceptible to…… this could be used in a discussion but not a conclusion from the study. In addition, you need to be specific, which diabetes you are talking about here, type 1 or type 2?

Response: Following the editor’s comment, we changed the last sentence of the conclusion.

Line 406-407: “The CaBP-9k mutation may be one of the predictive factors for high-risk individuals in the type 1 diabetes.”

7-all figures have the figure name appeared both on the top and upper part of the figure, omit the upper title.

Response: Following the reviewer’s comment, we deleted the upper title in all figures.

8- In all figures, all statistical differences were represented alphabetically which is misleading in such complex figures, it will look better by using standard symbols like *, **, *** etc.

Response: Following the reviewer’s comment, we changed all statistical differences from alphabetical to standard symbols like *, #, + in all figures.

9- If possible write the under each scale on figure the size of the scale.

Response: Following the reviewer’s comment, we added the size of the scale in all pictures.

10-on figure 6c: you put a line on the top of three bars and denoted a which p<0.05 vs naïve, why the line covers the three bars then?

Response: We apologize for the error lining in figure 6c. We removed the line in the revised figure.

11- the title overall could be improved: The protective role of Calbindin-D9k on endoplasmic reticulum induced beta cell death.

Response: Following the reviewer’s comment, we changed title from “The role of calbindin-D9k in beta cell against endoplasmic reticulum stress induced cell death” to “The protective role of calbindin-D9k on endoplasmic reticulum induced beta cell death” in revised manuscript.
